# REASONING-BASED PERSONALIZED GENERATION FOR USERS WITH SPARSE DATA

## ABSTRACT

Large Language Model (LLM) personalization holds great promise for tailoring responses by leveraging personal context and history. However, real-world users usually possess sparse interaction histories with limited personal context, such as cold-start users in social platforms and newly registered customers in online E-commerce platforms, compromising the LLM-based personalized generation. To address this challenge, we introduce GRASPER (**Gra**ph-based **S**parse **Pe**rsonalized **R**easoning), a novel framework for enhancing personalized text generation under sparse context. GRASPER first augments user context by predicting items that the user would likely interact with in the future. With reasoning alignment, it then generates texts for these interactions to enrich the augmented context. In the end, it generates personalized outputs conditioned on both the real and synthetic histories, ensuring alignment with user style and preferences. Extensive experiments on three benchmark personalized generation datasets show that GRASPER achieves significant performance gain, substantially improving personalization in sparse user context settings.

## 1 INTRODUCTION

Personalized Large Language Models (LLMs) have recently garnered significant attention (Salemi et al., 2023; Tsai et al., 2024) due to their various downstream applications in search, recommendation, and conversational agents (Yoganarasimhan, 2019; Qian et al., 2014; Shumanov & Johnson, 2021). By retrieving relevant personal context from user history, LLMs can produce outputs that are tailored to the given user's personal preferences and enhance overall satisfaction and quality.

The core of LLM personalization lies in retrieving personal context, typically derived from a user's history. However, most existing approaches emphasize textual histories (Salemi et al., 2023; 2024a). While useful, these histories are often sparse and limited, which severely constrains personalization for long-tail users due to insufficient context. For example, in e-commerce platforms and social

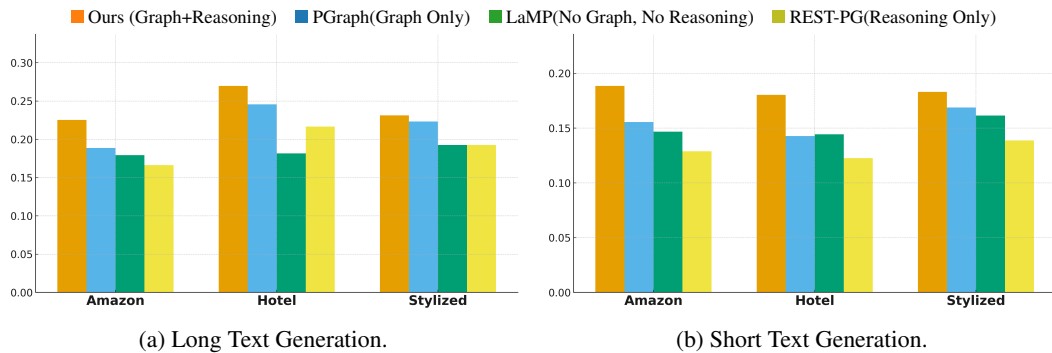

(a) Long Text Generation.        (b) Short Text Generation.

Figure 1: Results comparing our approach across two fundamental tasks and across datasets. Legends use descriptive labels: Ours (Graph + Reasoning), PGraph (Graph Only), LaMP (No Graph, No Reasoning), and REST-PG (Reasoning Only). GRASPER achieves over 10% gains, on average, across datasets.

networks, more than 95% of users tend to be cold-start (Au et al., 2025; Ni & McAuley, 2018). Recent studies show that incorporating auxiliary product reviews from other users—naturally represented as user-item graphs that connect users, items, and their interactions—can effectively improve model performance in settings such as recommendation and text generation (Au et al., 2025; Wang et al., 2025). By leveraging such graph-based histories, language models are able to generate more accurate and contextually rich personalized responses.

Despite these advancements, how to effectively leverage diverse data sources, such as graphs, to improve data sparsity in personalized LLMs remains underexplored. While prior work (Au et al., 2025) focused on retrieving existing textual histories for target items, it ignores the potential of more complex, structural information in the user-item interaction graphs. Such rich structural data could provide crucial signals for users with otherwise sparse textual histories, as shown in recent works (Wang et al., 2025). Furthermore, the integration of varied contextual information often lacks a dedicated reasoning phase before generation. We argue that this reasoning step is crucial as the diversity and volume of retrieved information requires the LLM to strategically synthesize and generate coherent and personalized outputs (Salemi et al., 2025).

To address these challenges, we propose GRASPER, a framework with two key stages: graph-based augmentation and reasoning-aligned generation. In the augmentation stage, we integrate a pretrained link predictor to enrich the sparse user context by simulating potential future interactions. In the generation stage, the personalized text generation model leverages both the augmented history of the user and existing texts of the target item to craft tailored responses. Crucially, reasoning is explicitly incorporated into both stages: during augmentation, reasoning guides the generation of features for the simulated edges to ensure alignment with user preferences; during text generation, reasoning enforces consistency between the synthesized history and the final personalized response (Salemi et al., 2025). Extensive experiments demonstrate that GRASPER substantially outperforms state-of-the-art baselines.

Our contribution can be summarized as follows:

- We introduce GRASPER, a novel framework that tackles sparsity in personalized text generation by combining graph learning and reasoning. The graph-based augmentation expands limited user histories with simulated texts of predicted interactions, while reasoning ensures coherence between augmented and original data.

- To the best of our knowledge, this is the first work to explicitly integrate reasoning into sparse personalization, enabling LLMs to cohesively synthesize augmented context and generate faithful, user-aligned outputs.

- We conduct extensive experiments on real-world datasets (Amazon (Ni & McAuley, 2018), Hotel (Kanouchi et al., 2020), and Stylized Feedback (Alhafni et al., 2024)), showing that GRASPER achieves, on average, over 10% improvement on state-of-the-art baselines for text generation, and 15% for rating prediction.

## 2 PROBLEM DEFINITION

Most existing works on personalized text generation assume that users have access to rich personal contexts, typically requiring more than ten historical text entries (Au et al., 2025; Salemi et al., 2023). However, this assumption does not align with real-world usage patterns, where user data often follows a long-tail distribution—over 95% of users have written fewer than two text entries (Ni & McAuley, 2018; Au et al., 2025). This data sparsity poses a significant challenge for personalization systems that rely heavily on individual user histories.

We argue that the naturally occurring bipartite user-item interaction graph, constructed from collective user-item histories, offers a valuable source of auxiliary context (Zhao et al., 2021b;a). It encodes implicit relationships between users and items through shared interactions and textual feedback, providing a structural foundation to infer user preferences even in low-resource settings.

Thus, to address the sparsity challenge and enrich the user context in such cases, we follow the setting in prior work (Au et al., 2025) and define the task of *graph-based personalized text generation*. In this task, the goal is to leverage both the sparse personal history of a user and the broader structural context encoded in the user-item graph to generate personalized text (e.g., reviews) for a target item.

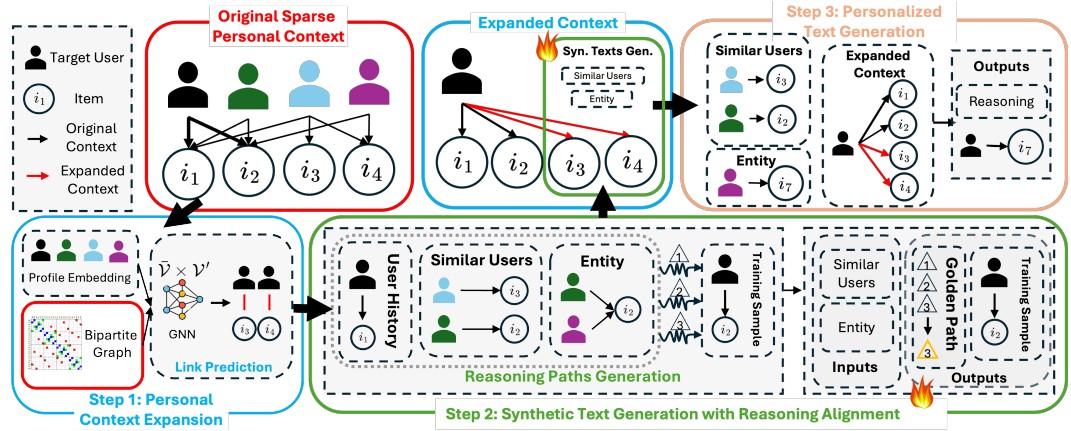

Figure 2: Overview of the proposed GRASPER. In **Step 1**, the personal context is enriched by leveraging the underlying graph structure to predict potential interactions. In **Step 2**, we generate synthetic reviews for the predicted interactions with aligned reasoning. In **Step 3**, the enhanced personal context that contains both the observed and simulated interactions enables the generation of more accurate and personalized text.

**Personalized Text Generation with Graph.** We formally define the task in this section. Let $\mathcal{U} = \{u_1, u_2, \ldots, u_n\}$ denote the set of $n$ users, and $\mathcal{I} = \{i_1, i_2, \ldots, i_m\}$ denote the set of $m$ target items (e.g., products, hotels). For each user $u \in \mathcal{U}$, we define their interaction history as a sequence

$$H_u = \big[(i_{u,1}, t_{u,1}), \ldots, (i_{u,k_u}, t_{u,k_u})\big],$$

where $k_u$ is the number of interactions of user $u$ (which may vary across users), $i_{u,\ell} \in \mathcal{I}$ is the $\ell$-th item interacted with by $u$, $t_{u,\ell} \in \mathcal{T}$ is the associated text, such as the review written by a customer or a comment written by a social media user, and $\mathcal{T}$ is the space of all possible text.

Following previous work (Au et al., 2025), we model the interactions as a bipartite graph $G = (\mathcal{U} \cup \mathcal{I}, \mathcal{E})$, where each edge $(u, i) \in \mathcal{E}$ exists if $(i, t) \in H_u$ for some text $t$ and user $u$.

The goal of personalized text generation is to learn a function:

$$f : \mathcal{U} \times \mathcal{H} \times \mathcal{I} \to \mathcal{T}$$

that generates a personalized text $t_{u,i^*}$ for a given user $u \in \mathcal{U}$, their history $H_u \in \mathcal{H}$, a new target item $i^* \in \mathcal{I}$ and the bipartite graph $G$:

$$t_{u,i^*} = f(u, H_u, i^*),$$

where $\mathcal{H}$ is the space of all possible user histories.

## 3 METHOD

As illustrated in Figure 2, GRASPER operates in three steps to address the challenge of personalization under sparse context: (**Step 1**) *Personal Context Expansion via Link Prediction*, which augments sparse histories with potential user–item interactions predicted from a graph model; (**Step 2**) *Synthetic Review Generation with Reasoning Alignment*, which integrates intermediate reasoning paths into the text generation process to ensure personalization alignment; and (**Step 3**) *Personalized Text Generation*, which produces personalized outputs conditioned on the augmented personal context.

### 3.1 STEP 1: PERSONAL CONTEXT EXPANSION

Personalization often suffers in the long-tail setting where users have limited history, as limited contexts constrain downstream LLM generation quality and further compromise personalization. To mitigate this limitation, we propose enriching sparse histories by predicting additional user–item interactions through graph-based link prediction. These predicted edges allow us to construct a more comprehensive profile for each user, which can then be leveraged in later reasoning alignment and generation stages. To simulate additional user–item interactions, we perform link prediction over the bipartite user–item interaction graph $G = (\mathcal{U} \cup \mathcal{I}, \mathcal{E})$ defined in Section 2.

**Training Graph-based Link Predictor.** Each node $v \in \mathcal{U} \cup \mathcal{I}$ is initialized with an embedding $\mathbf{h}_v^{(0)} \in \mathbb{R}^d$ derived from textual features using a pretrained encoder:

$$\mathbf{h}_v^{(0)} = \begin{cases} \text{Enc}(T_u), & v = u \in \mathcal{U}, \\ \text{Enc}(R_i), & v = i \in \mathcal{I}, \end{cases} \tag{1}$$

where $\text{Enc}(\cdot)$ denotes a sentence encoder mapping text to a $d$-dimensional embedding, $T_u$ is the concatenated string of the sequence of text in user $u$'s profile $T_u = \text{Concat}[t_{u,1}, ..., t_{u,k_u}]$, and $R_i = \{ t_{u,i} \mid (u,i) \in E \}$ is the set of texts aggregated for item $i$. Enc can be text embedding model (Reimers & Gurevych, 2019). Since style is a crucial element for personalization compared to content itself (Zhang et al., 2025), we instantiate Enc as a style-aware encoder that disentangles stylistic elements from content following Wegmann et al. (2022).

With the initialized node features, we adopt GraphSAGE (Hamilton et al., 2018) for inductive graph encoding. At each layer $l$, a node $v$ updates its representation by first aggregating its neighbors $\mathcal{N}(v)$ and then combining this aggregation with its own embedding:

$$\mathbf{m}_v^{(l)} = \text{AGG}^{(l)}\Big( \big\{ \mathbf{h}_u^{(l-1)} : u \in \mathcal{N}(v) \big\} \Big), \quad \mathbf{h}_v^{(l)} = \sigma\Big( \mathbf{W}^{(l)} \big[ \mathbf{h}_v^{(l-1)} \parallel \mathbf{m}_v^{(l)} \big] \Big) \tag{2}$$

where $\text{AGG}^{(l)}$ is a permutation-invariant mean aggregator, $\mathbf{W}^{(l)}$ is a learnable projection matrix, $\parallel$ denotes concatenation, and $\sigma$ is a non-linear activation function (ReLU in our implementation). After $L$ layers, we obtain the final node embedding $\mathbf{z}_v = \mathbf{h}_v^{(L)}$. In conjunction with the style-aware initialization, $\mathbf{z}_v$ captures both preference and stylistic elements of the personalization target.

To estimate the likelihood of a new interaction between a user $u$ and an item $i$, we apply a decoder over their embeddings. Specifically, we compute a score with a multi-layer perception (MLP) and turn it into a probability as:

$$s_{u,i^*} = \text{MLP}([\mathbf{z}_u \parallel \mathbf{z}_{i^*}]), \qquad \hat{y}_{u,i^*} = \text{Sigmoid}(s_{u,i^*}). \tag{3}$$

The link predictor is trained with binary cross-entropy (BCE) loss, where observed interactions are positives and uniformly sampled non-interacted pairs are negatives:

$$\mathcal{L}_{\text{link}} = - \sum_{(u,i) \in E^+} \log \sigma(s_{u,i}) \; - \sum_{(u,i) \in E^-} \log\big(1 - \sigma(s_{u,i})\big), \tag{4}$$

with $E^+ = E$ and $E^-$ constructed via negative sampling (Zhang & Chen, 2018).

**Inference and Profile Augmentation.** At inference, for each sparse user $u$, we score all candidate items $i \in \mathcal{I} \setminus \{ i : (u,i) \in \mathcal{E} \}$, rank them by $s_{u,i^*}$, and select the top-$K$ predictions. Let $\mathcal{I}_u^K = \{ i_{u,1}, \ldots, i_{u,K} \}$ denote this top-$K$ set. For each $i \in \mathcal{I}_u^K$, we generate a synthetic text (as detailed in Step 2) to approximate how $u$ might interact with it, and these texts are appended to $u$'s history. The result is an augmented user profile that incorporates both observed and predicted interactions:

$$\tilde{H}_u = H_u \cup \{ \tilde{t}_{u, i_{u,k}} : k = 1, \ldots, K \}. \tag{5}$$

where $\tilde{t}_{u,i}$ explicitly denotes the synthetically generated text for user $u$ on item $i$. Importantly, these predicted edges are used only locally for the given user $u$. For example, an edge $(u,i)$ generated for user $u$ does not affect another user $u'$ through the shared item $i$. Thus, we are not reconstructing a full new bipartite graph but enriching each user profile independently for downstream personalization.

## 3.2 STEP 2: SYNTHETIC TEXT GENERATION WITH REASONING ALIGNMENT

The augmented profiles from Step 1 yield predicted candidate items, but directly generating texts from these signals risks propagating noise or stylistic mismatch. To address this, we design a reasoning-based synthetic text generation process that integrates explicit reasoning before producing final outputs.

**Synthetic Text Generation Setup.** For a target user $u$, let $\mathcal{S}_u$ denote the set of similar users identified from the user–item graph $G$, and let $\mathcal{I}_u$ be the set of items inferred in Step 1 that $u$ is likely to interact with. $\mathcal{S}_u$ is obtained by calculating the cosine similarity between the node embeddings $\mathbf{z}_v$. We select the top 3 similar users to construct $H_{S_u}$. For each item $i \in \mathcal{I}_u$, we aim to generate a synthetic text $\tilde{y}_{u,i}$ that reflects $u$'s style and preferences. The generation process conditions on three sources of input:

$$x = \{H_u, H_{\mathcal{S}_u}, P_{u,i}\}, \tag{6}$$

where $H_u$ are past texts of $u$, $H_{\mathcal{S}_u}$ are texts written by similar users, and $P_{u,i}$ are peer texts associated with item $i$. Following Au et al. (2025), $P_{u,i}$ is constructed by ranking all texts associated with item $i$ using the BM25 retrieval model (Robertson et al., 1995) and selecting the top 4 most relevant entries, where relevance is measured by semantic similarity to the input query.

**Reasoning Path Generation.** We use a language model $\mathcal{M}$ to produce intermediate rationales that explain why $u$ might write for item $i$ in a certain way. Formally, a reasoning path $\mathcal{Z}$ is an intermediate textual explanation conditioned on $x$, such as *"User $u$ tends to prefer lightweight laptops; similar users highlighted battery life for item $i$; hence the review should emphasize portability and battery."* During training, for each text entry $t_{u,j}$, we let $x = \{H_u \setminus t_{u,j}, H_{\mathcal{S}_u}, P_{u,j}\}$. We then obtain a set of candidate reasoning paths by sampling with a formatted prompt $\phi(x, t_{u,j})$(The prompt formulation is supplied in Appendix G) that includes the input and the expected output

$$\mathcal{Z}^{(r)} = \mathcal{M}(\phi(x, t_{u,j})), \quad r = 1, 2, \ldots, R. \tag{7}$$

We sample the candidate reasoning paths because not all candidate reasoning paths are equally reliable. We select a *golden* reasoning path $\mathcal{Z}^*$ that best aligns with ground-truth outputs by maximizing task performance under an evaluation metric $\Omega$, which we take the average of the ROUGE and METEOR scores (see Appendix C.4 for more details on the evaluation metrics)

$$\mathcal{Z}^* = \arg\max_{\mathcal{Z}} \Omega\big(t'_{u,j}, t_{u,j}\big), \quad \text{where } t'^{(r)}_{u,j} = \mathcal{M}(\xi(x, \mathcal{Z}^{(r)})), \tag{8}$$

with $t'^{(k)}_{u,j}$ denoting the generated synthetic text, and $\xi$ a prompt-construction function combining $x$ and $\mathcal{Z}$. The specification of $\xi$ is supplied in Appendix G.

**Reasoning Alignment.** Finally, the model is fine-tuned to jointly generate the selected reasoning path $\mathcal{Z}^*$ and text $t_{u,j}$:

$$\mathcal{L}_{\text{gen}} = \text{CE}(\mathcal{M}(\rho(x)), t_{u,j}), \tag{9}$$

where $\rho$ is the prompt formatting function that is designed to generate both the reasoning and the output (The prompt formulation is supplied in Appendix G). $\mathcal{M}(\rho(x))$ is trained to output $\mathcal{Z}^*$ followed by the text. It enables the model to leverage noisy augmentation while staying faithful to user-specific style and preferences. Let the fine-tuned model be $\mathcal{M}'$, for a predicted item $i \in \mathcal{I}_u^K$, the final synthetic text $\tilde{t}_{u,i}$ is then generated with $\mathcal{M}'(x) \setminus \mathcal{Z}'$ where $\mathcal{Z}'$ is the generated reasoning.

### 3.3 STEP 3: PERSONALIZED TEXT GENERATION

The final stage reuses the reasoning aligned language model from Step 2, which defines a mapping from user histories, similar-user signals, and candidate items to reasoning paths and synthetic texts. In Step 3, we employ the same function for the personalization task.

Let $\mathcal{I}_u$ be the set of candidate items inferred in Step 1. For each $i \in \mathcal{I}_u$, Step 2 already learns to map from the user's profile history, texts written by similar users, and peer texts associated with the predicted target item, as shown in Equation (6). Note that for personalized text generation for a specific target item $i^*$, we have the same input formulation $x^* = [\tilde{H}_u, H_{S_u}, P_{u,i^*}]$ where insteadwof profile history, $\tilde{H}_u$ is the augmented history obtained from Step 1. Since the reasoning–generation function is shared, we reuse the fine-tuned model $\mathcal{M}'$ in Step 2 for the target text generation. Given the input $x^*$, the fine-tuned model $\mathcal{M}'$ produces both a reasoning path $z^*$ and a personalized text $\hat{t}_{u,i^*}$:

$$\mathcal{M}'(x^*) = [z^* \parallel \hat{t}_{u,i^*}]. \tag{10}$$

The final prediction strips away the reasoning tokens:

$$\hat{t}_{u,i^*} = \mathcal{M}'(x^*) \setminus z^*. \tag{11}$$

Table 1: Results comparing the proposed approach called GRASPER to state-of-the-art methods across 3 different tasks on the Amazon Review benchmark.

| Task | Metric | LLM | GRASPER (ours) | PGraph | LaMP | REST-PG |
|------|--------|-----|---------|--------|------|---------|
| **LONG TEXT GEN.** | ROUGE-1 ↑ | *4o-mini* | **0.219** | 0.189 | 0.171 | N/A |
| | | *LlaMA3* | **0.215** | 0.178 | 0.173 | 0.165 |
| | ROUGE-L ↑ | *4o-mini* | **0.170** | 0.130 | 0.117 | N/A |
| | | *LlaMA3* | **0.171** | 0.129 | 0.129 | 0.109 |
| | METEOR ↑ | *4o-mini* | 0.182 | **0.196** | 0.176 | N/A |
| | | *LlaMA3* | **0.178** | 0.151 | 0.138 | 0.122 |
| | LLM-as-a-Judge ↑ | *4o-mini* | **0.421** | 0.389 | 0.328 | N/A |
| | | *LlaMA3* | **0.337** | 0.297 | 0.277 | 0.269 |
| **SHORT TEXT GEN.** | ROUGE-1 ↑ | *4o-mini* | **0.178** | 0.115 | 0.108 | N/A |
| | | *LlaMA* | **0.155** | 0.131 | 0.124 | 0.089 |
| | ROUGE-L ↑ | *4o-mini* | **0.174** | 0.112 | 0.105 | N/A |
| | | *LlaMA3* | **0.153** | 0.125 | 0.118 | 0.081 |
| | METEOR ↑ | *4o-mini* | **0.162** | 0.099 | 0.091 | N/A |
| | | *LlaMA3* | **0.142** | 0.125 | 0.117 | 0.113 |
| | LLM-as-a-Judge ↑ | *4o-mini* | **0.406** | 0.353 | 0.334 | N/A |
| | | *LlaMA3* | **0.304** | 0.241 | 0.228 | 0.232 |
| **RATING PRED.** (Recommendation) | RMSE ↓ | *4o-mini* | **0.33** | 0.38 | 0.34 | N/A |
| | | *LlaMA3* | **0.32** | 0.76 | 0.72 | 0.65 |
| | MAE ↓ | *4o-mini* | **0.34** | 0.73 | 0.70 | N/A |
| | | *LlaMA3* | **0.31** | 0.34 | 0.31 | 0.46 |

## 4 EXPERIMENTS

### 4.1 DATASETS AND METRICS

We follow the experiment setup in prior works (Au et al., 2025), which consists of three datasets: Amazon Review (Ni & McAuley, 2018), Hotel Review (Kanouchi et al., 2020), and Stylized Feedbacks (Alhafni et al., 2024). We cover three personalization tasks: long text generation, short text generation, and rating prediction. For long text generation, a title will be given to guide the generation, and for short text generation, a paragraph will be given for summarization.

For both long and short text generation tasks, we adopt widely used lexical overlap metrics, including ROUGE-1, ROUGE-L, and METEOR, following prior work (Au et al., 2025; Kumar et al., 2024; Salemi et al., 2023). To complement these surface-level metrics, we further incorporate LLM-as-a-Judge evaluation (Salemi et al., 2025; Liu et al., 2023), where a strong language model provides comparative assessments of personalization. Additional details on LLM-as-a-Judge can be found in Appendix B. For rating prediction, we use Root Mean Squared Error (RMSE) and Mean Average Error (MAE). Further details on the metrics and experimental setup can be found in Appendix C.

### 4.2 BASELINES

We benchmark against three state-of-the-art personalization baselines. **LaMP** (Salemi et al., 2023) conditions on a user's past writing via prompts but uses no graph learning or reasoning. **PGraphRAG** (Au et al., 2025) (i.e., PGraph) employs graph-based retrieval augmented generation with BM25 for personalization but lacks reasoning or fine-tuning. **REST-PG** (Salemi et al., 2025) models user preferences through reasoning paths with iterative fine-tuning, but it does not employ any context expansion. Expanded baseline descriptions can be found in Appendix C. We evaluate LaMP and PGraph with LLaMA-3-8b-instruct and GPT-4o mini; REST-PG is implemented with LLaMA-3-8b-instruct only due to its fine-tuning requirement.

### 4.3 MAIN RESULTS

We report the experimental results for the Amazon Reviews, Hotel Experience, and Stylized Feedback datasets in Tables 1 to 3, respectively. These results evaluate our approach across three tasks—long text generation, short text generation, and ordinal classification—and compare its performance to the state-of-the-art personalization baselines. Additional results are provided in Appendix E.

Table 2: Performance Metrics for Hotel Experience Generation.

| Task | Metric | LLM | GRASPER (ours) | PGraph | LaMP | REST-PG |
|------|--------|-----|----------------|--------|------|---------|
| **LONG TEXT GEN.** | ROUGE-1 ↑ | *4o-mini* | 0.257 | **0.263** | 0.221 | N/A |
| | | *LlaMA3* | 0.258 | **0.263** | 0.199 | 0.221 |
| | ROUGE-L ↑ | *4o-mini* | **0.163** | 0.152 | 0.135 | N/A |
| | | *LlaMA3* | **0.168** | 0.157 | 0.129 | 0.131 |
| | METEOR ↑ | *4o-mini* | 0.165 | 0.184 | 0.164 | N/A |
| | | *LlaMA3* | 0.165 | **0.191** | 0.152 | 0.145 |
| | LLM-as-a-Judge ↑ | *4o-mini* | **0.500** | 0.414 | 0.300 | N/A |
| | | *LlaMA3* | **0.488** | 0.372 | 0.246 | 0.369 |
| **SHORT TEXT GEN.** | ROUGE-1 ↑ | *4o-mini* | **0.135** | 0.112 | 0.108 | N/A |
| | | *LlaMA3* | **0.147** | 0.127 | 0.126 | 0.100 |
| | ROUGE-L ↑ | *4o-mini* | **0.128** | 0.111 | 0.104 | N/A |
| | | *LlaMA3* | **0.140** | 0.118 | 0.117 | 0.091 |
| | METEOR ↑ | *4o-mini* | **0.120** | 0.081 | 0.075 | N/A |
| | | *LlaMA3* | **0.130** | 0.102 | 0.106 | 0.084 |
| | LLM-as-a-Judge ↑ | *4o-mini* | **0.469** | 0.360 | 0.346 | N/A |
| | | *LlaMA3* | **0.304** | 0.224 | 0.228 | 0.215 |
| **RATING PRED.** | RMSE ↓ | *4o-mini* | 0.660 | **0.328** | 0.340 | N/A |
| | | *LlaMA3* | **0.322** | 0.347 | 0.326 | 0.335 |
| | MAE ↓ | *4o-mini* | 0.356 | **0.336** | 0.700 | N/A |
| | | *LlaMA3* | **0.520** | 0.724 | 0.680 | 0.642 |

Table 3: Performance Metrics for Stylized Feedback Generation.

| Task | Metric | LLM | GRASPER (ours) | PGraph | LaMP | REST-PG |
|------|--------|-----|----------------|--------|------|---------|
| **LONG TEXT GEN.** | ROUGE-1 ↑ | *4o-mini* | **0.214** | 0.185 | 0.187 | N/A |
| | | *LlaMA3* | **0.235** | 0.217 | 0.186 | 0.189 |
| | ROUGE-L ↑ | *4o-mini* | **0.147** | 0.123 | 0.123 | N/A |
| | | *LlaMA3* | **0.175** | 0.158 | 0.134 | 0.127 |
| | METEOR ↑ | *4o-mini* | 0.178 | **0.183** | 0.189 | N/A |
| | | *LlaMA3* | 0.175 | **0.178** | 0.177 | 0.173 |
| | LLM-as-a-Judge ↑ | *4o-mini* | **0.423** | 0.399 | 0.318 | N/A |
| | | *LlaMA3* | **0.340** | 0.340 | 0.273 | 0.281 |
| **SHORT TEXT GEN.** | ROUGE-1 ↑ | *4o-mini* | **0.137** | 0.122 | 0.113 | N/A |
| | | *LlaMA3* | **0.160** | 0.149 | 0.140 | 0.097 |
| | ROUGE-L ↑ | *4o-mini* | **0.133** | 0.118 | 0.109 | N/A |
| | | *LlaMA3* | **0.157** | 0.142 | 0.134 | 0.091 |
| | METEOR ↑ | *4o-mini* | **0.144** | 0.104 | 0.096 | N/A |
| | | *LlaMA3* | 0.131 | **0.142** | 0.136 | 0.112 |
| | LLM-as-a-Judge ↑ | *4o-mini* | **0.395** | 0.343 | 0.331 | N/A |
| | | *LlaMA3* | **0.284** | 0.242 | 0.236 | 0.255 |
| **RATING PRED.** | RMSE ↓ | *4o-mini* | **0.637** | 0.673 | 0.667 | N/A |
| | | *LlaMA3* | **0.684** | 0.724 | 0.680 | 0.678 |
| | MAE ↓ | *4o-mini* | **0.332** | 0.347 | 0.344 | N/A |
| | | *LlaMA3* | **0.337** | 0.347 | 0.327 | 0.326 |

Overall, our method consistently outperforms the baselines across the datasets and tasks. For long text generation, we observe significant improvements in ROUGE-1, ROUGE-L, and METEOR scores, demonstrating the model's ability to generate more accurate and contextually relevant outputs through reasoning enhanced retrieval and generation. Notably, the Amazon Reviews dataset shows the largest performance gains. This can be attributed to it having the fewest average degree of 1.68 comparing to 2.12 and 2.42 for the Hotel Experience dataset and Sylized Feedback dataset (see Appendix C.1). The increased sparsity leads to better performance gain with the proposed method. For ordinal classification, our approach achieves lower RMSE and MAE compared to the baselines, indicating better alignment with user rating tendencies. Furthermore, we find that the relative advantage of our method becomes more pronounced under LLM-as-a-Judge evaluation. Compared with conventional textual similarity metrics, which primarily capture surface-level overlap, LLM-as-a-Judge better aligns with human preference when assessing personalization (Salemi et al., 2025; Liu et al., 2023). This is because personalization often extends beyond literal similarity to reflect nuanced aspects such as style, tone, and contextual coherence. The gains observed in this setting demonstrate that GRASPER, by jointly leveraging context expansion and reasoning alignment, is able to generalize user-specific stylistic patterns more effectively, leading to outputs that are not only accurate but also more faithful to individual user preferences.

Table 4: Module ablation studies for Amazon Reviews dataset with the Llama3 backbone. GRASPER-ft is an ablation without fine-tuning, and GRASPER-r-ft is without reasoning and fine-tuning.

| Task | Metric | GRASPER | GRASPER-ft | GRASPER-r-ft | PGraph | LaMP | REST-PG |
|------|--------|---------|------------|--------------|--------|------|---------|
| **Text Generation** | ROUGE-1 | **0.215** | 0.175 | 0.182 | 0.178 | 0.173 | 0.165 |
| | ROUGE-L | **0.171** | 0.121 | 0.125 | 0.129 | 0.129 | 0.009 |
| | METEOR | 0.178 | 0.188 | **0.200** | 0.151 | 0.138 | 0.122 |
| **Title Generation** | ROUGE-1 | **0.155** | 0.100 | 0.118 | 0.131 | 0.124 | 0.112 |
| | ROUGE-L | **0.153** | 0.098 | 0.117 | 0.125 | 0.118 | 0.077 |
| | METEOR | **0.142** | 0.085 | 0.111 | 0.125 | 0.117 | 0.113 |
| **Rating Prediction** | RMSE | **0.32** | 0.53 | 0.52 | 0.52 | 0.72 | 0.65 |
| | MAE | **0.31** | 0.39 | 0.36 | 0.34 | **0.31** | 0.46 |

## 4.4 ABLATION STUDIES

We conduct extensive ablations on GRASPER, examining its components to validate the framework's effectiveness. We further analyze the key hyperparameter K (number of predicted items) along with supporting theory. Results using additional language models are presented in Appendix E.

### 4.4.1 MODEL VARIANTS

In Table 4, we ablate GRASPER with additional variants to demonstrate the effectiveness of its two main contributions: personal context expansion and reasoning alignment. GRASPER-ft removes the fine-tuning for reasoning, which means it will only include the reasoning prompt. GRASPER-r-ft further removes the reasoning process and lets the model directly generate the final output with the input as specified in Equation (6). The full model (GRASPER) consistently outperforms its reduced counterparts, confirming that each component is indispensable to the framework.

**Effect of Context Expansion.** Personal context expansion provides additional evidence for personalization, but without proper reasoning, the augmented context can introduce noise. This is reflected in GRASPER-ft-r, which relies solely on link prediction without reasoning or fine-tuning. These results indicate that context expansion alone is insufficient and may even hurt performance if not paired with reasoning alignment. The effect of noise-induced bias is extensively discussed in Appendix F.

**Effect of Reasoning Alignment.** Reasoning alignment ensures that the augmented context contributes in a way that matches user preferences and task requirements. Comparing GRASPER-ft (with reasoning but no fine-tuning) to GRASPER shows that reasoning alignment improves performance across metrics. Additionally, reasoning alignment without additional context (as in REST-PG) also underperforms, since the model lacks sufficient personalized evidence to reason over.

The results demonstrate that context expansion and reasoning alignment are complementary. With only context expansion (GRASPER-r-ft), the model introduces noise and degrades performance. With only reasoning, even when aligned with finetuning (e.g., REST-PG), the model has nothing substantial to reason over. Only by combining both can GRASPER achieve better personalization across text generation and rating prediction.

### 4.4.2 HYPERPARAMETER ANALYSIS

The hyperparameter $K$ as defined in Equation (5) controls how many candidate items we add via the link predictor when augmenting a user's personal context. In our main experiments, we fix K = 2 for efficiency, but the design of our method allows more effective and robust use with larger values of K. By comparison, PGraph expands context by retrieving K nearest reviews to the query embedding. However, because it lacks explicit reasoning alignment, its performance drops as K grows — the extra retrieved context introduces more noise than benefit (as seen in Table 5).

Our theoretical analysis (detailed in Appendix F) shows that this phenomenon follows a bias–variance trade-off. Adding more synthetic context reduces variance (helping especially sparse users with few real samples), but also introduces bias from preference mismatch. Without alignment, the bias term grows with K, limiting the benefit. However, with reasoning alignment, the mismatch is effectively shrunk: the bias is reduced, so larger K values become safe and beneficial. This explains why in experiments our method continues to improve as K increases, while PGraph plateaus or even degrades. Full mathematical details of this trade-off are provided in the Proposition 1.

Table 5: Sensitivity of GRASPER and PGraph to the neighborhood size $k$ on the Amazon Reviews dataset with the Llama3 backbone. Best in **bold**, second-best underlined.

| Task | Metric | GRASPER | | | | PGraph | | | |
|---|---|---|---|---|---|---|---|---|---|
| | | $K=1$ | $K=2$ | $K=3$ | $K=4$ | $K=1$ | $K=2$ | $K=3$ | $K=4$ |
| **Long Text Gen.** | ROUGE-1 | **0.226** | 0.215 | 0.222 | 0.225 | 0.160 | 0.169 | 0.169 | **0.173** |
| | ROUGE-L | 0.172 | 0.171 | 0.173 | **0.177** | 0.121 | **0.125** | 0.124 | 0.124 |
| | METEOR | 0.184 | 0.178 | 0.189 | **0.196** | 0.125 | 0.138 | 0.149 | **0.150** |
| **Short Text Gen.** | ROUGE-1 | 0.168 | 0.155 | 0.168 | **0.195** | **0.128** | 0.123 | 0.112 | 0.125 |
| | ROUGE-L | 0.173 | 0.153 | 0.165 | **0.192** | **0.121** | 0.118 | 0.107 | 0.119 |
| | METEOR | 0.160 | 0.142 | 0.169 | **0.185** | 0.123 | **0.128** | 0.116 | 0.117 |

| Input | GraSPeR | PGraph | Ground Truth |
|---|---|---|---|
| Title: Disappointing. Profile: {} | The spray bottles broke in the first use. They were not worth the money. | Disappointment. | Didn't hold up. One broke apart first time used. Ended up buying different spray bottles. |
| Title: Great Option! Profile: {} | This is a great option! Love this product. It's gentle and leaves my skin feeling clean and soft. I would definitely recommend this to anyone who wants a gentle facial scrub that smells great. | This product is an excellent choice, it's really effective and leaves my skin feeling so smooth | This is a great option for an exfoliating cleanser! Love that there is an approachable option that is of a decent quality and isn't excessively costly. |
| Title: Nice. Profile: {} | I like it. It's comfortable and easy to use. | Looks good, nice and soft. | I like this - it does seem well made and it's just easy to wear. |

Figure 3: Case study with three examples. The matching green, blue, and yellow boxes show matching semantics or expression. The red box shows misalignment against the ground truth.

## 4.5 CASE STUDY

To better illustrate how GRASPER generates more faithful and personalized outputs, we present a case study in Figure 3. We compare outputs from GRASPER and PGraph against the ground truth under different input titles. As highlighted by the colored boxes, GRASPER consistently captures key semantics and stylistic expressions that align with the reference. For instance, in the first example, GRASPER generates "spray bottles broke in the first use," which mirrors both the semantics ("first use") and specific product mentioned ("spray bottles") in the ground truth. In contrast, PGraph only outputs a vague summary ("Disappointment") without grounding in the product context.

In the second example, GRASPER reproduces stylistic markers such as "great option" and "Love". Although PGraph's output "excellent choice" also captures the semantic meaning, it diverges from the ground truth in word choice. Finally, in the third example, both GRASPER and the ground truth emphasize short, colloquial phrasing ("I like it / I like this"). GRASPER also correctly matches the emphasis on usability ("easy to use / easy to wear"), while PGraph generates more general, less faithful wording ("Looks good, nice and soft"), deviating from the intended expression. These qualitative examples support our quantitative findings: context expansion and reasoning alignment together enable GRASPER to preserve fine-grained semantics and stylistic fidelity, while methods that rely only on raw augmentation often produce generic or misaligned outputs.

## 5 CONCLUSION

In this work, we proposed GRASPER, a reasoning-based framework for personalized text generation under sparse user contexts. By combining graph-based context expansion with explicit reasoning alignment, our method effectively enriches limited personal histories while ensuring generated outputs remain faithful to user style and preferences. Extensive experiments across datasets in different domains demonstrate that GRASPER significantly outperforms strong baselines.

Our findings highlight that context expansion and reasoning alignment are complementary: expansion alone risks introducing noise, while reasoning without sufficient context lacks grounding. Together, they enable models to better capture fine-grained semantics and stylistic fidelity, even for long-tail users with minimal histories. We believe this reasoning-enhanced paradigm opens promising directions for future research in large language model personalization, especially in real-world scenarios where data sparsity is the norm.

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

## A RELATED WORK

**LLM Personalization.** Personalization in LLMs has recently garnered significant attention Salemi et al. (2024b); Zhang et al. (2025) due to its potential to improve various downstream applications, including search, recommendation, and conversational agents Yoganarasimhan (2019); Qian et al. (2014); Shumanov & Johnson (2021). Salemi et al. (2023) introduced the LaMP benchmark, which comprises seven datasets designed to evaluate personalization in language models by incorporating personal context into downstream generation and classification tasks. Building on this, Salemi et al. (2024a) explores optimization strategies for personalization by improving the retriever's ability to select relevant personal context. Furthermore, Kumar et al. (2024) extends personalization research to the domain of long-form text generation. In addition, to address the issue of long-tail data sparsity in the personal context histories, Au et al. (2025) proposes to augment the personal context with user-centric graphs, retrieving relevant histories from other users.

**LLM Reasoning and Planning.** Reasoning in LLM encourages the language model to think and plan before generation, leading to more coherent and accurate outputs Li et al.. Chain-of-Thought (CoT) prompting Wei et al. (2023) was first proposed to elicit reasoning capabilities of language models by prompting the model to generate a series of intermediate steps that lead to a final answer. Various methods have extended the CoT prompting to address its deficiencies. For example, Self-Consistency Wang et al. (2023) samples multiple reasoning paths and selects the most consistent answer, mitigating the impact of occasional reasoning errors. Additionally, Tree-of-Thoughts(ToT) Yao et al. (2023) allow LLMs to explore multiple reasoning paths in a tree-like structure, performing deliberate lookahead and backtracking to make more informed decisions. Graph-of-Thoughts(GoT) Besta et al. (2024) further generalizes the ToT by modeling the reasoning processes as arbitrary graphs.

Recently, several works have explored reasoning in personalization. REST-PG Salemi et al. (2025) employs self-training on LLM reasoning paths to improve the personalization. Luo et al. (2025) explored reinforced reasoning for personalization by incorporating and refining a hierarchical reasoning thought template to guide the reasoning process. Additionally, Kim et al. (2025) explored reasoning-level personalization that aligns model's reasoning process with a user's personalized logic. Several works have also explored reasoning to enhance personalized recommendations Lyu et al. (2024); Bismay et al. (2024); Yang et al. (2023) beyond traditional item-based recommendations.

## B LLM-AS-A-JUDGE

Traditionally, in the prior personalization benchmarks (Au et al., 2025; Kumar et al., 2024; Salemi et al., 2023), personalized text generation has been evaluated with lexical overlap metrics such as ROUGE (Lin, 2004). However, it has been shown that such metrics may fail to capture the semantic nuances and stylistic alignment in personalization. Thus, we adopt the LLM-as-a-Judge prompt from prior works on personalized text generation Salemi et al. (2025), which is designed based on the evaluation paradigm introduced in Liu et al. (2023). Our prompt is introduced as follows.

> **LLM-as-a-Judge**
>
> Please compare the generated text to the reference text based on how well they match and/or are similar.
>
> Scoring Scale:
> 1 – Strongly disagree
> 2 – Disagree
> 3 – Somewhat disagree
> 4 – Neither agree nor disagree
> 5 – Somewhat agree
> 6 – Agree
> 7 – Strongly agree
>
>
> Content to Evaluate:
> Reference Text (Ground Truth): {target_text}
> Generated Text: {generated_text}
>
>
> Provide only the numeric score (1–7).

We use GPT-4 as the judge LLM, and report the normalized score (0.1-0.7) in our main experiment table. Salemi et al. (2025) designed additional experiments to validate the effectiveness of the LLM-as-a-Judge evaluation. First, they conduct a human evaluation comparing 100 model outputs and find that the LLM-as-a-Judge scores agree with human preference in 73% of cases, with a Pearson correlation of 0.46. Second, they design a controlled perturbation study by randomly replacing a portion of the personalized contexts with unrelated ones. The LLM-as-a-Judge scores decrease linearly as the perturbation rate increases, showing that the evaluator is sensitive to mismatched personalization.

While no automatic metric can fully replicate human evaluation for personalization—since the "true" judge of style and preference is the original user—LLM-as-a-Judge provides a scalable and semantically meaningful proxy. In our setting, it enables consistent evaluation across sparse and noisy contexts, capturing personalization quality beyond what lexical metrics can measure.

## C EXPERIMENTAL SETUP

### C.1 DATASET STATISTICS

| Dataset | Train Size | Validation Size | Test Size |
|---|---|---|---|
| User-Product Review | 20,000 | 2,500 | 2,500 |
| Stylized Feedback | 20,000 | 2,500 | 2,500 |
| Hotel Experiences | 9,000 | 2,500 | 2,500 |

Table 6: Dataset split sizes across training, validation, and test sets for the four domains.

We provide the dataset statistics in this section. In Table 6, we give the train/validation/test split statistics for the datasets. It is worth noting that the Hotel Experience dataset is a smaller dataset with a smaller training set, leading to the more inconsistent performance that we presented in the Experiment section. In Table 7, we introduce the task statistics for Long Text Generation, Short Text Generation, and Ordinal Classification. The datasets are constructed to reflect the real-world distribution (Au et al., 2025), which results in the sparse profiles as shown in the Average Profile Size. GRASPER achieves more consistent and significant performance gain in scenarios where the output length is shorter, such as Short Text Generation and the User-Product Review (Amazon dataset), as longer text implicitly gives more context for text generation.

| Task | Type | Avg. Input Length | Avg. Output Length | Avg. Profile Size | # Classes |
|------|------|-------------------|--------------------|--------------------|-----------|
| User-Product Review Generation | Long Text Generation | $3.754 \pm 2.71$ | $47.90 \pm 19.28$ | $1.05 \pm 0.31$ | - |
| Hotel Experiences Generation | Long Text Generation | $4.29 \pm 2.57$ | $76.26 \pm 22.39$ | $1.14 \pm 0.61$ | - |
| Stylized Feedback Generation | Long Text Generation | $3.35 \pm 2.02$ | $51.80 \pm 20.07$ | $1.09 \pm 0.47$ | - |
| User-Product Review Title Generation | Short Text Generation | $30.34 \pm 37.95$ | $7.02 \pm 1.14$ | $1.05 \pm 0.31$ | - |
| Hotel Experiences Summary Generation | Short Text Generation | $90.40 \pm 99.17$ | $7.64 \pm 0.92$ | $1.14 \pm 0.61$ | - |
| Stylized Feedback Title Generation | Short Text Generation | $37.42 \pm 38.17$ | $7.16 \pm 1.11$ | $1.09 \pm 0.47$ | - |
| User-Product Review Ratings | Ordinal Classification | $34.10 \pm 38.66$ | - | $1.05 \pm 0.31$ | 5 |
| Hotel Experiences Ratings | Ordinal Classification | $94.69 \pm 99.62$ | - | $1.14 \pm 0.61$ | 5 |
| Stylized Feedback Ratings | Ordinal Classification | $40.77 \pm 38.69$ | - | $1.09 \pm 0.47$ | 5 |

Table 7: Data statistics for the PGraphRAG Benchmark across the four datasets. For each task, we report the average input and output lengths (in words), measured on the test set using BM25-based retrieval with GPT. The average profile size indicates the number of reviews per user used for personalization.

## C.2 DATASETS

We evaluate our approach on three benchmark datasets introduced in prior research Au et al. (2025). These datasets cover diverse domains and graph structures, enabling us to assess the effectiveness of our method.

**Amazon Review.** The Amazon Review dataset is constructed from the Amazon Review 2023 corpus (Hou et al., 2024). We build a user-item interaction graph where nodes represent users and products, and edges indicate review interactions between them.

**Hotel Experience.** The Hotel Experience dataset is collected from the Datafiniti Hotel Reviews dataset (Au et al., 2025). It contains user-hotel interaction data, where edges denote users' stays at hotels and are annotated with textual reviews.

**Stylized Feedback Review.** The Stylized Feedback Review dataset is derived from the Datafiniti Grammar and Online Product dataset (Au et al., 2025). It focuses on generating stylistic and domain-specific feedback from user-product interactions. This dataset emphasizes linguistic diversity and style adaptation.

## C.3 TASKS

Here we present an extended discussion on the tasks that we used to evaluate GRASPER: Long Text Generation, Short Text Generation, and Ordinal Classification.

**Long Text Generation.** The long text generation task focuses on producing detailed user reviews given a review title and the user's profile. The objective is to generate coherent and contextually relevant review text that aligns with the user's preferences. This task evaluates the model's capability for generating high-quality, personalized text.

**Short Text Generation.** The short text generation task involves generating concise product titles or summaries given a user review. The challenge lies in distilling a longer text into a shorter title. This task assesses the model's ability to distill information from highly personalized user context.

**Ordinal Classification.** The ordinal classification task aims to predict the rating score a user would assign to a product based on the title and review text. This task is particularly challenging because of varying rating behaviors; for example, some users might write critical reviews yet still assign high scores. This task is designed to evaluate the model's ability to capture subtle patterns in user preferences and rating tendencies.

## C.4 METRICS

**Text Generation.** For both long and short text generation tasks, we adopt widely used lexical overlap metrics, including ROUGE-1 and ROUGE-L, following prior work Au et al. (2025). These metrics capture n-gram and subsequence overlaps between the generated output and ground-truth references. To complement these surface-level measures, we further incorporate LLM-as-a-Judge evaluation, where a strong language model provides comparative assessments of personalization and accuracy. We design the prompt based on prior studies which has been validated with human evaluators on the task of personalization (Salemi et al., 2025). The prompt for LLM-as-a-Judge evaluation is provided in Appendix B.

**Ordinal Classification.** For the ordinal classification task, we evaluate rating prediction using Root Mean Squared Error (RMSE) and Mean Absolute Error (MAE). RMSE penalizes large deviations more heavily, highlighting extreme mispredictions, while MAE measures the average magnitude of prediction errors.

## D PSEUDO CODE

---

**Algorithm 1** GRASPER — Training

---

**Require:** Bipartite graph $G = (U \cup I, E)$; user histories $\{H_u\}$; item reviews $\{R_i\}$; encoder $\mathrm{Enc}(\cdot)$; base LLM $\mathcal{M}$; hyperparameters: $K$ (items to augment), $k_{\mathrm{sim}}$ (similar users), $k_{\mathrm{peer}}$ (peer texts)
**Ensure:** Trained link predictor (GraphSAGE + MLP), fine-tuned LLM $M'$

  1: **// Step 1: Personal Context Expansion**
  2: **for** each node $v \in U \cup I$ **do**
  3:     **if** $v$ is user $u$ **then**
  4:         $h_v^{(0)} \leftarrow \mathrm{Enc}(\mathrm{concat}(H_u))$
  5:     **end if**
  6:     **if** $v$ is item $i$ **then**
  7:         $h_v^{(0)} \leftarrow \mathrm{Enc}(R_i)$
  8:     **end if**
  9: **end for**
 10: **for** $\ell = 1$ to $L$ **do**                                       ▷ GraphSAGE layers
 11:     $m_v^{(\ell)} \leftarrow \mathbf{AGG}_\ell\big(\{\, h_u^{(\ell-1)} : u \in \mathcal{N}(v)\,\}\big)$
 12:     $h_v^{(\ell)} \leftarrow \mathrm{ReLU}\big(W_\ell[\, h_v^{(\ell-1)} \| m_v^{(\ell)}\,]\big)$
 13: **end for**
 14: $z_v \leftarrow h_v^{(L)}$ for all $v$
 15: Score edges with $s(u,i) = \mathrm{MLP}([z_u \| z_i])$, $\hat{y}(u,i) = \sigma(s(u,i))$
 16: Optimize BCE with negative sampling to train (GraphSAGE+MLP) $\rightarrow$ Link Predictor

 17: **// Step 2: Synthetic Review Generation with Reasoning Alignment**
 18: **for** each training user $u$ **do**
 19:     $\mathcal{S}_u \leftarrow \mathrm{TopK}_{k_{\mathrm{sim}}}\big(\cos(z_u, z.)\big)$
 20:     $H_{\mathcal{S}_u} \leftarrow \{\text{reviews from users in } \mathcal{S}_u\}$
 21:     **for** each observed pair $(u, j)$ **do**
 22:         $P_{u,j} \leftarrow \mathrm{BM25\_TopK}_{k_{\mathrm{peer}}}(\text{reviews of } j)$
 23:         $x \leftarrow \{\, H_u \setminus \{t_{u,j}\},\ H_{\mathcal{S}_u},\ P_{u,j}\,\}$
 24:         Sample candidate reasoning paths $\{Z^{(1)}, \ldots, Z^{(K)}\} \sim M$ with prompt $\phi(x)$
 25:         $Z^\star \leftarrow \arg\max_Z\ \Omega\big(M(\xi(x, Z)), t_{u,j}\big)$     ▷ $\Omega$: dev metric (e.g., ROUGE/METEOR)
 26:         Update $M$ by minimizing $\mathrm{CrossEntropy}\big(M(\rho(x)),\ [Z^\star \| t_{u,j}]\big)$
 27:     **end for**
 28: **end for**
 29: $M' \leftarrow M$
 30: **return** (Link Predictor), $M'$

---

---

**Algorithm 2** GRASPER — Inference

---

**Require:** Trained (GraphSAGE+MLP), $M'$; graph $G$; $\{H_u\}$, $\{R_i\}$; $K$, $k_{\text{sim}}$, $k_{\text{peer}}$; target $(u, i^\star)$
**Ensure:** Personalized review $\hat{t}_{u,i^\star}$

 1: **// Step 1: Personal context expansion**
 2: Initialize $h_v^{(0)}$ with $\text{Enc}(\cdot)$; run GraphSAGE to obtain $z_v$ for all $v$
 3: Rank items $i \in I \setminus \{i : (u,i) \in E\}$ by $s(u,i)$; let $\mathcal{I}_u^K \leftarrow \text{TopK}_K$
 4: $\mathcal{S}_u \leftarrow \text{TopK}_{k_{\text{sim}}}\big(\cos(z_u, z.)\big)$
 5: **for** each $i \in \mathcal{I}_u^K$ **do**
 6: $\quad P_{u,i} \leftarrow \text{BM25\_TopK}_{k_{\text{peer}}}(\text{reviews of } i)$
 7: $\quad x_i \leftarrow \{ H_u,\ H_{\mathcal{S}_u},\ P_{u,i} \}$
 8: $\quad [z_i' \,\|\, \tilde{t}_{u,i}] \leftarrow M'(x_i)$ $\qquad\qquad\qquad\qquad$ ▷ reasoning + synthetic review
 9: **end for**
10: $\tilde{H}_u \leftarrow H_u \cup \{\tilde{t}_{u,i} : i \in \mathcal{I}_u^K\}$

11: **// Step 2: Final personalized generation for target item**
12: $P_{u,i^\star} \leftarrow \text{BM25\_TopK}_{k_{\text{peer}}}(\text{reviews of } i^\star)$
13: $x^\star \leftarrow \{ \tilde{H}_u,\ H_{\mathcal{S}_u},\ P_{u,i^\star} \}$
14: $[z^\star \,\|\, \hat{t}_{u,i^\star}] \leftarrow M'(x^\star)$
15: **return** $\hat{t}_{u,i^\star}$

---

Table 8: Backbone ablation on Amazon Reviews dataset for GRASPER using open-source (Llama 3, Gemma 2) and proprietary (GPT-4o mini, GPT-4.1) backbones. Metrics for text/title generation are higher-is-better; for rating prediction, lower-is-better.

| Category | Backbone | Method | Text Generation | | | | Title Generation | | | | Rating Prediction | |
|---|---|---|---|---|---|---|---|---|---|---|---|---|
| | | | R-1 | R-L | MET | LJ | R-1 | R-L | MET | LJ | RMSE | MAE |
| Open-Source | Llama 3 | GRASPER | **0.215** | **0.171** | **0.178** | **0.337** | 0.155 | **0.153** | 0.142 | **0.304** | **0.32** | **0.31** |
| | | PGraph | 0.178 | 0.129 | 0.151 | 0.297 | **0.178** | 0.129 | **0.151** | 0.241 | 0.76 | 0.34 |
| | Gemma 2 | GRASPER | **0.160** | **0.119** | **0.121** | **0.326** | **0.127** | **0.123** | 0.122 | **0.329** | **0.52** | **0.53** |
| | | PGraph | 0.155 | 0.119 | 0.117 | 0.316 | 0.098 | 0.093 | **0.124** | 0.297 | 0.88 | 0.42 |
| Proprietary | GPT-4o mini | GRASPER | **0.219** | **0.170** | 0.182 | **0.421** | **0.178** | **0.174** | **0.162** | **0.406** | **0.33** | **0.34** |
| | | PGraph | 0.189 | 0.130 | **0.196** | 0.389 | 0.115 | 0.112 | 0.099 | 0.353 | 0.38 | 0.73 |
| | GPT-4.1 | GRASPER | **0.221** | **0.176** | 0.181 | **0.433** | **0.151** | **0.150** | **0.166** | **0.401** | **0.31** | **0.32** |
| | | PGraph | 0.185 | 0.128 | **0.191** | 0.403 | 0.107 | 0.103 | 0.122 | 0.346 | 0.38 | 0.70 |

# E  ADDITONAL EXPERIMENT RESULTS

## E.1  LANGUAGE MODEL VARIANTS

In Table 8, we compare GRASPER across different backbone models, covering both open-source (Llama 3, Gemma 2) and proprietary (GPT-4o mini, GPT-4.1) variants. This setup allows us to test whether the improvements of GRASPER depend on a particular language model family or extend across architectures with varying sizes and training pipelines.

For open-source models, GRASPER consistently improves over the baseline PGraph across all metrics. With Llama 3, GRASPER achieves a clear gain in text generation and rating prediction. Gemma-2, though smaller in scale, still benefits from our framework, showing improved semantic quality on LLM-as-a-Judge metric. These results suggest that GRASPER effectively enhances smaller open-source models, making them more competitive for personalization tasks.

When applied to proprietary models, the improvements remain consistent. On GPT-4o mini, GRASPER outperforms the baseline in text generation and especially in LLM-as-a-Judge, demonstrating better alignment with human preferences. GPT-4.1 mini, the more advanced backbone, also benefits: GRASPER achieves the highest score across metrics, indicating strong personalization quality even when starting from a more powerful model.

Table 9: Link Prediction Performance across Different Datasets

| Dataset | MRR | Hits@1 | Hits@5 | Hits@10 |
|---------|-----|--------|--------|---------|
| Amazon | 0.531 | 0.415 | 0.659 | 0.760 |
| Hotel | 0.324 | 0.210 | 0.446 | 0.546 |
| Feedbacks | 0.275 | 0.178 | 0.394 | 0.477 |

Overall, the results confirm that GRASPER is robust to the choice of language model backbone. Gains are observed consistently across both open-source and proprietary families. Importantly, improvements in LLM-as-a-Judge are more significant, underscoring that our framework aligns better with human preference. This robustness highlights GRASPER 's practicality, as it can be flexibly deployed in various settings with different backbone models.

## E.2 LINK PREDICTION NOISE AND ROBUSTNESS

Although GRASPER achieves strong improvements on personalized text generation—particularly in sparse-user settings—the link prediction module inevitably introduces a degree of noise due to imperfect edge predictions. It is therefore important to assess both the quality of the predicted user–item links and the robustness of the downstream reasoning-based personalization to such noise.

Table 9 reports the standalone performance of the link prediction module across all datasets. The module demonstrates consistently strong ranking metrics, indicating its ability to recover meaningful user–item affinities even under sparse supervision. Nonetheless, some level of incorrect or low-confidence predictions is unavoidable. To study whether such noise impacts the final generation quality, we further conduct an analysis on the Amazon test set by partitioning examples into two groups: the top 50% and bottom 50% based on their link-prediction confidence scores.

Table 10: Comparison of Text Generation Scores by Link Prediction Performance

| Group | ROUGE-L Mean | METEOR Mean |
|-------|--------------|-------------|
| Bottom 50% link pred scores | 0.587982 | 0.555729 |
| Top 50% link pred scores | 0.605974 | 0.569973 |

As shown in Table 10, the generation quality of the low-confidence group remains comparable to that of the high-confidence group across ROUGE and METEOR metrics. This suggests that even when some retrieved neighbors originate from noisy edges, the reasoning module is able to filter, contextualize, and extract stylistically relevant information from the neighborhood. Overall, these results indicate that GRASPER is robust to moderate imperfections in link prediction and can effectively leverage the noisy-but-useful relational signals present in sparse user–item graphs.

## E.3 PERSONALIZATION SPARSITY ROBUSTNESS

To further examine how GRASPER behaves under different levels of personalization sparsity, we partition users in the test split by the number of real historical reviews available: users with 0 reviews (cold-start), 1 review, and 2+ reviews. This allows us to isolate how much GRASPER depends on explicit user history versus the contextual and relational reasoning signals introduced by our framework. As shown in Table 11, GRASPER demonstrates strong robustness across all sparsity levels and consistently outperforms the PGraph baseline. Notably, our model achieves meaningful improvements even in the cold-start setting. This behavior arises because, even when a user has no prior reviews, GRASPER can still leverage contextual cues provided at inference time, including the review title, partial user-written text, or product description, to retrieve relevant neighbors and construct a personalized reasoning path. By contrast, prior personalization methods such as PGraph depend primarily on embedding-based retrieval, which is significantly less effective when a user lacks a profile or has only one review.

| Sparsity Level | GRASPER ROUGE-L | PGraph ROUGE-L | GRASPER ROUGE-1 | PGraph ROUGE-1 |
|---|---|---|---|---|
| 0 historical reviews (cold-start) | **0.160** | 0.125 | **0.204** | 0.183 |
| 1 historical review | **0.170** | 0.137 | **0.225** | 0.212 |
| 2+ historical reviews | **0.186** | 0.149 | **0.296** | 0.262 |

Table 11: Performance under different sparsity levels of user history.

### E.4 UTILITY OF REASONING PATH SELECTION

In this section, we further explore the utility of the reasoning path selection as introduced in Eq. 7. We analyze the ranked reasoning paths produced by the scoring metric $\Omega$. The distribution of candidate scores, shown in Table 12, reveals clear separation among candidate paths, indicating that their quality varies and that the proposed selection mechanism is necessary for GRASPER to identify the most coherent and stylistically aligned reasoning trace.

| **Rank** | 1 (lowest) | 2 | 3 | 4 | 5 (highest) |
|---|---|---|---|---|---|
| **Score** | 0.3943 | 0.4465 | 0.4709 | 0.4964 | 0.5137 |

Table 12: Score distribution of ranked reasoning paths produced by $\Omega$ (Eq. 8).

These results indicate that while reasoning traces cannot be directly evaluated in isolation, the model benefits substantially from the ranked reasoning guidance.

## F THEORETICAL ANALYSIS OF THE BIAS-VARIANCE TRADE-OFF IN GRASPER

As detailed in Section 4.4.2, the hyperparameter $K$ determines the number of predicted items. In Table 5, we demonstrate that GRASPER, with the reasoning alignment, can more reliably utilize the additional retrieved context compared to other baselines with retrieval. We hypothesize the behavior corresponds to the bias-variance trade-off theory, where the reasoning serves as a regularization trick that can offset the trade-off and allow the variance reduction without bias increase. Note that in the below we use k instead of K to represent the number of predicted/synthetic.

**Proposition 1** (Bias–Variance trade-off)**.** *Let $\theta_u \in \mathbb{R}^d$ denote the user's latent style vector. We observe $n$ real samples $x_j = \theta_u + \varepsilon_j$ with $\mathbb{E}[\varepsilon_j] = 0$, $\mathrm{Var}(\varepsilon_j) = \sigma^2 I$, and $k$ synthetic samples $\tilde{x}_\ell = \theta_u + \Delta + \tilde{\varepsilon}_\ell$ with $\mathbb{E}[\tilde{\varepsilon}_\ell] = 0$, $\mathrm{Var}(\tilde{\varepsilon}_\ell) = \tilde{\sigma}^2 I$, where $\Delta \in \mathbb{R}^d$ is a fixed (unknown) bias. Consider the pooled estimator*

$$\hat{\theta}_u = \frac{1}{n+k}\Big(\sum_{j=1}^{n} x_j + \sum_{\ell=1}^{k} \tilde{x}_\ell\Big).$$

*Then the (per-coordinate) mean squared error is*

$$\mathrm{MSE}(k) = \underbrace{\frac{n\sigma^2 + k\tilde{\sigma}^2}{(n+k)^2}}_{variance} + \underbrace{\Big(\frac{k}{n+k}\Big)^2 \|\Delta\|^2}_{bias^2/d\ (per\text{-}dim)}.$$

*In the equal-noise case $\tilde{\sigma}^2 = \sigma^2$, this simplifies to*

$$\mathrm{MSE}(k) = \frac{\sigma^2}{n+k} + \Big(\frac{k}{n+k}\Big)^2 \|\Delta\|^2.$$

*Sketch.* $\mathbb{E}[\hat{\theta}_u] = \theta_u + \frac{k}{n+k}\Delta$, so the squared bias per dimension is $\big(\frac{k}{n+k}\big)^2 \|\Delta\|^2$ (treating $\sigma^2$ as per-dimension noise). Since samples are independent with isotropic noise, $\mathrm{Var}(\hat{\theta}_u) = \frac{n\sigma^2 + k\tilde{\sigma}^2}{(n+k)^2} I$. Add variance and bias$^2$ to obtain the expression. For $\tilde{\sigma}^2 = \sigma^2$, write $\mathrm{MSE}(t) = \frac{\sigma^2}{n}(1-t) + \|\Delta\|^2 t^2$, differentiate w.r.t. $t$, set to zero, and solve. ∎

*Remark* 1 (Effect of Reasoning Alignment). If reasoning alignment attenuates the preference mismatch to $\Delta_{\text{RA}} = \beta\Delta$ with $\beta \in (0, 1)$, then

$$\text{MSE}_{\text{RA}}(k) \;=\; \frac{\sigma^2}{n+k} \;+\; \left(\frac{k}{n+k}\right)^2 \beta^2 \|\Delta\|^2,$$

so the minimum achievable error decreases and the optimal fraction $t_{\text{RA}}^\star = \frac{\sigma^2}{2n\beta^2\|\Delta\|^2}$ increases, i.e., *alignment lets you safely use larger $k$.*

*Remark* 2 (Sparse Users Benefit More). $t^\star$ scales as $1/n$: when $n$ is small (sparse users), the variance term dominates and the optimal augmentation fraction is larger. Thus augmentation disproportionately helps sparse users by reducing variance, while reasoning alignment curbs the bias induced by $\Delta$.

## G GRASPER PROMPTS

In this section, we supply the prompts we used in GRASPER. $\phi$ is used in Equation (7) where the prompt is used to elicit candidate reasoning paths. $\xi$ is used in Equation (8), where the prompt is used to generate the final answer given the input and reasoning to evaluate the candidate paths. Lastly, $\rho$ is used in Equation (9) where it structures the final input for personalized text generation.

---

**Reasoning Paths Generation ($\phi$)**

System: You are a personalized review generation assistant that generates high-quality reviews based on user history and context.

Given profile which contains past documents written by the same person (might be empty), documents written by users that have similar writing style, reviews on the target product, and reasoning.

User's own profile: {history_reviews_str}
Similar profiles: {neighbor_reviews_str}
Product Reviews: {product_reviews_str)}

Based on the above information, provide a detailed reasoning path that explains how we can arrive at the expected output. Consider:
1. User's Writing Style: Analyze their typical review length, tone, and language patterns.
2. User's Preferences: What aspects of products do they typically focus on or value?
3. Product Information: What are the commonly mentioned features, pros, and cons from other reviews?
Do not limit the reasoning to the above points. You can use your own knowledge to reason about the user's review. It is important to make sure that you only talk about information from the profile while considering the expected output in the reasoning process. You cannot directly copy or mention anything about the expected output. The expected output is only used to determine the reasoning process and how profile can affect the expected output.

Provide your reasoning that leads to the following expected review on the target product from the user:
Expected Output:
Title: "target_review['title']"
Text: "target_review['text']"
Rating: target_review['rating']

As mentioned before, you cannot directly copy or mention anything about the expected output. The expected output is only used to determine the reasoning process. Do not mention the expected output in your reasoning. Your reasoning should only analyze the profile and the other reviews.

Output your reasoning in a single paragraph. Do not output anything else.

---

Your reasoning:

---

**Reasoning Paths Evaluation ($\xi$)**

System: You are a personalized review evaluation assistant that judges whether the generated reasoning and review are consistent with the user's style and product context.

Given a profile containing past documents written by the same person (may be empty), documents from users with similar writing style, reviews on the target product, and a reasoning trace, you will evaluate and refine the review text.

User's own profile: {history_reviews_str}

Similar profiles: {neighbor_reviews_str}

Product Reviews: {product_reviews_str)}

Reasoning: {reasoning_str}

Based on the above information, evaluate how well the provided review text follows the reasoning and user profile. Consider:
1. Faithfulness to the reasoning: Does the review follow the logical path outlined in the reasoning?
2. Stylistic alignment: Does the review reflect the user's writing style and preferences?
3. Product grounding: Is the review consistent with the product reviews and features mentioned?

Do not copy directly from the reasoning or profiles. Your task is to provide a short evaluation and, if needed, produce a refined review text.

Provide your output strictly in the format:
Evaluation: <evaluation>. Review text: <Review text>

Do not output anything else.

Review text: {review_text}

---

**Text Generation ($\rho$)**

System: You are a personalized review generation assistant that generates high-quality reviews based on user history and context.

Given a profile containing past documents written by the same person (may be empty), documents written by users with similar writing style, and reviews on the target product.

User's own profile: {history_reviews_str}

Similar profiles: {neighbor_reviews_str}

Product Reviews: {product_reviews_str)}

Reason and generate a review title/review text based on the following review text/review title. Use the format:
Reasoning: <reasoning>. Review title/text: <Review title/text>.

Do not output anything else.

Review text/Review title: {review_text} / {review_title}

---

# H NOTATIONS

To facilitate readability, we summarize the main mathematical symbols and notations used throughout the paper in Table 13, which serves as a quick reference to clarify definitions of variables, functions, and operators appearing in the main text.

Table 13: Summary of key notations used throughout the paper.

| Symbol | Type / Shape | Description |
|---|---|---|
| $G = (\mathcal{U} \cup \mathcal{I}, \mathcal{E})$ | graph | Graph with user set $\mathcal{U}$, items $\mathcal{I}$, edges $\mathcal{E}$ |
| $H_u$ | set of texts | Observed history (reviews) of user $u$ |
| $R_i$ | set of texts | Reviews associated with item $i$ |
| $\text{Enc}(\cdot)$ | text $\rightarrow \mathbb{R}^d$ | Text encoder (e.g. SentenceTransformers) |
| $h_v^{(\ell)}$ | $\mathbb{R}^d$ | Node representation at layer $\ell$ (GraphSAGE) |
| $z_v = h_v^{(L)}$ | $\mathbb{R}^d$ | Final node embedding for node $v$ |
| $\mathcal{N}(v)$ | set of nodes | Neighborhood of node $v$ |
| $\mathbf{AGG}_\ell(\cdot)$ | operator | Neighborhood aggregator at layer $\ell$ |
| $\mathcal{M}$ | LLM | Base language model |
| $\mathcal{M}'$ | LLM | Fine-tuned LLM used for inference |
| $s_{u,i}$ | $\mathbb{R}$ | Link score from decoder for user $u$ and item $i$ |
| $K$ | integer | # predicted items to augment per user |
| $k_{\text{sim}}$ | integer | # similar users retrieved for $u$ |
| $k_{\text{peer}}$ | integer | # peer texts (BM25) per item |
| $\mathcal{S}_u$ | set of users | Top-$k_{\text{sim}}$ similar users to $u$ |
| $P_{u,i}$ | set of texts | Top-$k_{\text{peer}}$ peer reviews for item $i$ |
| $\mathcal{I}_u^K$ | set of items | Top-$K$ predicted items for user $u$ by $s(u,i)$ |
| $t_{u,i}$ | text | Ground-truth review by $u$ for item $i$ (observed) |
| $\tilde{t}_{u,i}$ | text | Synthetic review for $(u,i)$ during expansion |
| $\hat{t}_{u,i^\star}$ | text | Final predicted personalized review for user $u$ on target item $i^\star$ |
| $\mathcal{Z}$ | text | A reasoning path |
| $\{Z^{(k)}\}_{k=1}^K$ | list of texts | $K$ candidate reasoning paths |
| $\phi(x, t_{u,j})$ | prompt | Prompt to elicit candidate reasoning paths given input and expected output |
| $\xi(x, Z)$ | prompt | Prompt that conditions generation on input $x$ and rationale $\mathcal{Z}$ to evaluate $\mathcal{Z}$ |
| $\rho(x)$ | prompt | Prompt that instructs model to output the reasoning path and the review |
| $\Omega(\cdot, \cdot)$ | metric | Evaluation metric for reasoning paths (e.g., ROUGE/METEOR) |