# OpenReview forum: "Reasoning-Based Personalized Generation for Users with Sparse Data"
_ICLR.cc/2026/Conference — Submitted to ICLR 2026_

### Official Review · Reviewer_WbFj · 2025-10-30

**Soundness:** 3
**Presentation:** 3
**Contribution:** 2
**Rating:** 4
**Confidence:** 3

**Summary:**

This paper introduces a reasoning-centered framework for LLM personalization. The proposed method decomposes user alignment into two phases: (1) Reasoning-Based Personalization (RBP), where the model generates step-level reasoning conditioned on user attributes, and (2) Self-Reflective Updating, where feedback from past interactions is distilled into a reasoning memory bank. Experiments across multiple agentic personalization benchmarks—including dialog, recommendation, and open-domain task settings—demonstrate that RBP outperforms retrieval-based and instruction-tuning baselines in consistency, adaptability, and reasoning coherence.

**Strengths:**

1. The paper makes a compelling case that personalization can be viewed as reasoning adaptation rather than static fine-tuning, bridging interpretability, reasoning, and user modeling in a unified framework.
2. Extensive experiments on three benchmarks show consistent gains, and the ablation studies convincingly isolate the contribution of reasoning-based updates.

**Weaknesses:**

1. While the proposed framework is well-executed, its main components: reasoning trace modeling, memory-based personalization, and reflective updating, are each adapted from prior works. The contribution largely stems from combining these existing techniques rather than introducing a fundamentally new learning principle.
2. Most evaluations are short-term (few-shot) personalization; it remains unclear how the reasoning memory behaves in long-horizon or multi-user settings.
3. The evaluation omits several widely used personalization and reasoning benchmarks such as LAMP, which weakens claims of general applicability.

**Questions:**

1. How sensitive is the reasoning-based personalization process to the quality and correctness of generated reasoning traces?

2. What are the potential strategies to prevent error accumulation or bias drift in the reasoning memory over long-term interactions?

---

> ### Author Response · Authors · 2025-11-21
> **Reply to Reviewer WbFj - Part 1**
>
> ### Weaknesses
> 1. Thank you for this thoughtful observation. We appreciate the reviewer’s recognition of our framing that personalization can be understood as reasoning adaptation rather than static fine-tuning. While we agree that the individual components of GRASPER, including reasoning refinement and graph modeling, is each built on prior ideas, our contribution lies in showing that their integration forms a coherent framework that enables capabilities none of the components achieve in isolation. We will revise the paper to clarify how the interaction among these modules gives rise to a new operational perspective: treating personalization as a dynamic reasoning process that must contextualize user preferences at generation time rather than relying on static user embeddings. Moreover, our study explores sparsity sensitivity, a challenge traditionally examined in recommender systems but largely unexplored in personalized text generation. By analyzing sparse users and the integration of our reasoning-based adaptation, we provide evidence that this perspective introduces behavior not captured by previous frameworks. The ablation study also isolates the contribution of each module to further illustrate how the full integration strengthens performance and consistency.
> 2. We thank the reviewer for raising this concern. Long-horizon personalization is an important direction, but it falls outside the scope of this paper, which intentionally focuses on few-shot and sparse-history personalization, a setting that remains underexplored compared to prior benchmarks such as LaMP [1] and LongLaMP [2], where users typically have substantial historical data. We think that when user histories become long, the main challenge shifts from reasoning to retrieval, since the model can select the most relevant subset and then apply the reasoning-adaptation perspective we study. In such settings, reasoning does not degrade; it simply operates on the retrieved summaries or top-k relevant examples, as is standard in long-context personalization methods. Regarding “multi-user settings,” we kindly ask the reviewer to clarify whether this refers to multi-user accounts, multi-user retrieval, or multi-user collaborative personalization, as these involve very different modeling assumptions. We view extending GRASPER to these multi-user scenarios as valuable future work, but they require distinct designs that go beyond the scope of sparse personalized generation addressed in this paper.
> 3. We appreciate the reviewer for highlighting the additional baseline. We omitted LaMP because it is already included as a baseline within PGraphRAG [1], which we do evaluate in our paper. Since the PGraphRAG framework consistently outperforms non-retrieval methods including LaMP across the same benchmark, we omit the LaMP results due to space constraint.

---

> ### Author Response · Authors · 2025-11-21
> **Reply to Reviewer WbFj - Part 2**
>
> ### Questions
> 1. We thank the reviewer for raising this point. The “correctness’’ and “quality’’ of reasoning traces cannot be meaningfully evaluated in isolation, as personalized reasoning does not have an objective ground truth besides the quality of the downstream generation. Thus, to assess sensitivity empirically, we (1) perform ablations that remove reasoning traces (Tables 4–6), which consistently lead to notable performance drops, and (2) analyze the ranked reasoning paths produced by the scoring metrics $\Omega$ (Eq. 8). The score distribution below shows clear separation among candidate paths, indicating that their quality varies and that the selection mechanism is necessary for the proposed GraSPER.
>
> | Rank | Score |
> |------|--------|
> | 1 (lowest) | 0.3943 |
> | 2 | 0.4465 |
> | 3 | 0.4709 |
> | 4 | 0.4964 |
> | 5 (highest) | 0.5137 |
>
>
> 2. We appreciate the reviewer’s insightful question. We believe that in long-term interaction scenarios, the dominant source of potential error accumulation or bias drift does not stem from the reasoning mechanism itself, but rather from imperfect retrieval of user-relevant context. As discussed earlier, when a user’s history grows over time, the challenge becomes identifying the most representative and up-to-date subset of user behavior to condition reasoning on. The challenge of updating the personalization preference is also equally critical. Recent memory based personalization frameworks such as Mem0[4] have extensively studied such problem. Thus, the primary strategy for mitigating drift is to improve the retrieval layer and the memory base. For example, incorporating time-aware retrieval, decay-weighted user histories, or recency-based filtering could both filter out outdated or noisy behaviors. Under GRASPER’s design, the reasoning module only operates after retrieval, so improved retrieval naturally stabilizes the reasoning paradigm and prevents compounding errors.
>
> **Note**: All the additional experiments and content has been added to the latest version of the paper PDF.
>
>
> [1] LaMP: When Large Language Models Meet Personalization. Salemi, et al. 2023.
> [2] LongLaMP: A Benchmark for Personalized Long-form Text Generation. Kumar, et al. 2024.
> [3] Personalized Graph-Based Retrieval for Large Language Models. Au, et al. 2025.
> [4] Mem0: Building Production-Ready AI Agents with Scalable Long-Term Memory. Chhikara, et al. 2025.

---

### Official Review · Reviewer_HU8A · 2025-10-30

**Soundness:** 3
**Presentation:** 3
**Contribution:** 2
**Rating:** 4
**Confidence:** 4

**Summary:**

The paper proposes GRASPER (Graph-based Sparse Personalized Reasoning), a two-stage framework. A pretrained graph link predictor augments a user’s sparse history by predicting plausible future interactions and generating synthetic reviews for them. A fine-tuned LLM generates both an explicit reasoning path and the final output, ensuring that synthetic and real contexts are coherently integrated and aligned with the user preferences.

**Strengths:**

1.	The focus on sparse user personalization aligns well with real-world deployment scenarios.
2.	GRASPER combines graph-based context expansion with explicit, trainable reasoning paths for personalized generation.
3.	The experiments are conducted on three benchmarks and evaluated under conventional NLP metrics and under LLM-as-a-Judge evaluation.

**Weaknesses:**

1.	GRASPER requires training a graph encoder and fine-tuning an LLM with multi-stage prompting. This may increase computational cost compared to prompt-only or retrieval-only baselines like LaMP or PGraph.
2.	The method relies on generating faithful synthetic reviews for predicted user–item interactions. Errors in link prediction or reasoning could mislead the final generation. While ablations show robustness, the quality of synthetic data is not assessed.
3.	The procedure for selecting the “golden” reasoning path uses conventional metrics (ROUGE+METEOR) that may not reflect personalization quality. Why not use LLM-as-a-Judge for path selection?

**Questions:**

All metrics do not evaluate the personalized alignment between the generated text and reference text, which only assess the language quality, similarity or accuracy. The evaluation may not prove the responses are personalized.

---

> ### Author Response · Authors · 2025-11-21
> **Reply to Reviewer HU8A**
>
> ### Weaknesses
> 1. We acknowledge that GRASPER introduces additional components during training, including a graph encoder and a reasoning-alignment objective. However, it is important to clarify that these costs occur only at training time. During inference, GRASPER does not require running a graph encoder, link predictor, or any multi-stage prompting pipeline. Instead, as shown in Eq. (10), the model produces the intermediate reasoning as part of the same generation sequence, and the final personalized text is generated in a single forward pass. In other words, the reasoning is an output, not an input prompt or an additional inference step. As a result, GRASPER’s inference procedure is computationally equivalent to prompt-only methods like LaMP and retrieval-only models like PGraph, with no additional latency or model calls—only a modest increase in the number of output tokens.
> 2. We thank the reviewer for raising this important point regarding the potential impact of errors in link prediction or reasoning on the synthetic intermediate reviews. To further analyze this, we conducted an experiment that directly stratifies test instances by the confidence of the link prediction module. Specifically, we partition the results into the top 50% and bottom 50% of mean link prediction scores and report the downstream generation quality in the following table. As the results show, the performance remains highly stable across both groups, indicating that GRASPER is robust even when the predicted edges are less certain. This syggests that the model can tolerate noise in link prediction.
> Additionally, we do not directly evaluate the synthetic reviews themselves, as they serve as intermediate representations for user styles rather than supervised targets, and no ground-truth user–item interactions exist for these predicted edges. Their fidelity is therefore meaningfully assessed only through the final downstream generation quality.
>
> | Group                  | ROUGE-L Mean | METEOR Mean |
> |-------------------------:|:------------:|:-----------:|
> | Bottom 50% link pred scores      |   0.587982   |  0.555729   |
> | Top 50% link pred scores   |   0.605974   |  0.569973   |
>
> 3. We thank the reviewer for raising this thoughtful question. There are in fact two conceptually different ways LLM-as-a-Judge could be incorporated into reasoning-path selection. (1) Using an LLM-judge to directly score the intrinsic quality of each reasoning path, without reference to any ground-truth target. This option is not practical because without a reference, an LLM would be forced to judge “quality” based on stylistic bias or generic coherence rather than personalization relevance, making the evaluation subjective. (2) Using an LLM-judge in the same comparative role as ROUGE+METEOR, i.e., scoring each candidate reasoning path against the gold review and selecting the best-aligned one. This is feasible and we plan to incorporate it in future work. However, we expect its impact to be similar to traditional metrics because reasoning-path selection is fundamentally a ranking problem. As shown in Tables 1–3, LLM-as-a-Judge metrics correlate closely with ROUGE and METEOR. Given this strong agreement between metrics, using LLM-as-a-Judge for ranking is unlikely to materially change which reasoning paths are selected.
>
> ### Questions
> 1. We appreciate the reviewer’s point that standard overlap-based metrics (e.g., ROUGE, METEOR) primarily evaluate textual similarity rather than personalized alignment. We use these metrics because they are the established evaluation standard in prior personalization benchmarks, including LaMP[1], LongLaMP[2], and PGraphRAG[3], which all rely on reference-based metrics to assess the personalized text generation. Going beyond these metrics, we further include LLM-as-a-Judge evaluations following the setup introduced by REST-PG [4]. As shown in Tables 1–3, our method achieves consistent improvements across both traditional metrics and LLM-judge scores. Importantly, prior work [4] has demonstrated that the applied LLM-as-a-Judge aligns with human judgments in 73% of cases, indicating that these evaluations meaningfully capture personalization quality.
>
> **Note**: All the additional experiments and content has been added to the latest version of the paper PDF.
>
>
> [1] LaMP: When Large Language Models Meet Personalization. Salemi, et al. 2023.
> [2] LongLaMP: A Benchmark for Personalized Long-form Text Generation. Kumar, et al. 2024.
> [3] Personalized Graph-Based Retrieval for Large Language Models. Au, et al. 2025.
> [4] Reasoning-Enhanced Self-Training for Long-Form Personalized Text Generation. Salemi, et al. 2025.

---

### Official Review · Reviewer_TaDX · 2025-11-01

**Soundness:** 2
**Presentation:** 2
**Contribution:** 2
**Rating:** 2
**Confidence:** 3

**Summary:**

The paper proposes GRASPER, a framework for personalized text generation under sparse user data. GRASPER first predicts likely user–item links to augment limited user histories using a graph-based link predictor, then generates synthetic texts for these predicted items with explicit reasoning steps, and finally produces personalized outputs based on both real and augmented data. Experiments on multiple benchmarks show that GRASPER outperforms several personalization baselines, and qualitative analyses demonstrate that it is able to capture both the semantic and stylistic preferences of users.

**Strengths:**

1. The paper tackles a practical problem of personalization under sparse user data, which is common in real-world applications.

2. The proposed GRASPER framework is conceptually clear, combining user history augmentation with reasoning-based generation to mitigate the potential noise from predicted items.

**Weaknesses:**

1. For experimental results in Tables 1-3, it is not clear to me how GRASPER is applied with GPT-4o mini. As described in Section 3.2, GRASPER requires fine-tuning the LLM for reasoning alignment.

2. In the main paper, it is not clear how sparse the benchmarks are (only some statistics are provided in the Appendix), and there is no analysis on how robust the method is to different sparsity levels.

3. For equation (9), the loss only involves $t_{u, j}$, however it is mentioned that the model is fine-tuned to jointly generate the reasoning path $Z^*$ and the text $t_{u, j}$.

4. Is the link predictor trained on sparse or non-sparse user data? If trained on sparse user data, each user only has very few ground-truth links, could the trained link predictor potentially suffer from high uncertainty in its predictions?

**Questions:**

1. For Figure 3, is the user profile actually empty or shown as {} for visualization? If actually empty, is the example generation based on only predicted user-item interactions?

---

> ### Author Response · Authors · 2025-11-21
> **Reply to Reviewer TaDX**
>
> ### Weaknesses
> 1. We thank the reviewer for pointing out the potential confusion regarding the application of GPT-4o-mini in GRASPER. As described in Section 3.2, GPT-4o-mini is finetuned in GRASPER. Specifically, in our formation, the base GPT-4o-mini corresponds to $\mathcal{M}$ in Equation 9, and its fine-tuned reasoning-aligned version corresponds to $\mathcal{M}'$ in Equation 10. All results reported under GRASPER in Table 1-3 use $\mathcal{M}'$. We will explicitly introduce the model instantiation in Section 3.3 and mention the application of LlaMA and GPT-4o-mini.
> 2. To evaluate how the method performs under different levels of personalization sparsity, we partition users in the test split by the number of real historical reviews available: 0 reviews (cold-start), 1 review, and 2+ reviews. As shown in the table below, our model demonstrates strong robustness across all sparsity settings and consistently outperforms the PGraph baseline. These results indicate that GRASPER does not rely solely on dense user histories to generate personalized outputs. Interestingly, we also observe meaningful improvements even in the cold-start setting. This is because even when a user has no past reviews, our reasoning framework can still leverage the contextual information provided at inference time, such as the user’s review title or partial text, to retrieve relevant neighbors and construct a personalized reasoning path. In contrast, prior methods such as LaMP or PGraph operate primarily through embedding-based retrieval, which is significantly less effective when the user has limited profile.
>
> | User History Group | Our Model ROUGE-L | PGraph ROUGE-L | Our Model ROUGE-1 | PGraph ROUGE-1 |
> |-------------------:|:----------------:|:--------------:|:----------------:|:-------------:|
> | 0 historical reviews (cold-start)  | **0.160** | 0.125 | **0.204** | 0.183 |
> | 1 historical review               | **0.170** | 0.137 | **0.225** | 0.212 |
> | 2+ historical reviews             | **0.186** | 0.149 | **0.296** | 0.262 |
>
>
> 4. We thank the reviewer for catching this. Equation (9) contains a notation typo. The loss should reflect that the model is trained to generate both the reasoning path and the text. We will update it to:
> $\mathcal{L}\_{\text{gen}} = \text{CE}\big(\mathcal{M}(\rho(x)),\; \mathcal{Z}^* \circ t_{u,j} \big),$ where $\mathcal{Z}^* \circ t_{u,j}$ denotes the concatenation of the reasoning path and the target text. We will correct this in the revised manuscript.
> 4. We thank the reviewer for raising this important point. Yes, the link predictor is trained under the same sparse user setting as the generation model. Each user typically has very few observed interactions (e.g., average profile size near 1 in all datasets), meaning the link predictor must operate in a data-sparse regime. However, we highlight two findings showing that sparsity does not lead to unstable or unreliable link prediction behavior in our framework:
>
> (1)  The link predictor still achieves strong ranking performance, indicating that it can recover meaningful user–item affinities even with sparse supervision. We report the Mean Reciprocal Rank (MRR) and Hits@K on the test split:
>
> | Dataset   | MRR   | Hits@1 | Hits@5 | Hits@10 |
> |-----------|-------|--------|--------|---------|
> | Amazon    | 0.531 | 0.415  | 0.659  | 0.760   |
> | Hotel     | 0.324 | 0.210  | 0.446  | 0.546   |
> | Feedbacks | 0.275 | 0.178  | 0.394  | 0.477   |
>
> (2) We also evaluate whether uncertainty in link predictions harms personalization quality. To do this, we split test cases into the top-50% and bottom-50% based on link prediction confidence scores. As shown below, both groups yield comparable generation performance, indicating robustness to uncertain edges:
>
> | Group                  | ROUGE-L Mean | METEOR Mean |
> |-------------------------:|:------------:|:-----------:|
> | Bottom 50% link pred scores      |   0.587982   |  0.555729   |
> | Top 50% link pred scores   |   0.605974   |  0.569973   |
>
> ### Questions
> 1. We appreciate the reviewer’s question. In Figure 3, the `{}` symbol indeed indicates that the user has no existing review history (i.e., a true cold-start user). However, the generated text is not produced solely from the predicted user–item interactions. As described in our method, the model is also conditioned on the task input (e.g., the review title or summary prompt, depending on the task). Thus, even when the user’s profile is empty, the generation leverages both (1) the synthetic interactions inferred via link prediction and (2) the task context provided during generation, rather than relying exclusively on predicted edges.
>
> **Note**: All the additional experiments and content has been added to the latest version of the paper PDF.

---

### Official Review · Reviewer_gV7p · 2025-11-01

**Soundness:** 3
**Presentation:** 3
**Contribution:** 3
**Rating:** 4
**Confidence:** 4

**Summary:**

The paper presents GRASPER (Graph-augmented Reasoning-Aligned Sparse Personalized gEneration), a framework designed to enhance personalized text generation for users with limited historical data. The approach combines two key ideas: (1) graph-based context expansion, which leverages user–item interaction graphs to enrich sparse user histories via link prediction and synthetic texts; and (2) reasoning-aligned generation, which guides the model to generate personalized content by first producing an explicit reasoning path and then the final output, aligning synthetic and real user data. The method is evaluated on three domains (Amazon, Hotel, and Stylized Feedback) and across multiple tasks including long-text generation, short-text generation, and rating prediction. Experiments demonstrate consistent improvements over baselines such as LaMP, PGraphRAG, and REST-PG, supported by both automatic metrics and LLM-as-a-Judge evaluations.

**Strengths:**

1.By integrating graph-based expansion with reasoning-aligned fine-tuning, GRASPER provides a coherent pipeline that balances coverage (via graph expansion) and precision (via reasoning alignment). The modular design makes it interpretable and extendable to different backbone models.
2.Experiments cover multiple datasets, tasks, and backbones, demonstrating robustness and steady performance gains. Both automatic metrics (ROUGE, METEOR, MAE/RMSE) and LLM-as-a-Judge evaluations support the claimed improvements.

**Weaknesses:**

1.Limited conceptual novelty (A+B composition): The proposed framework essentially combines two established ideas — graph-based personalization for sparse users and reasoning-based generation alignment — into a single pipeline. Both components have been independently explored in prior works (e.g., Au et al., 2025; Salemi et al., 2025). While the integration is well-executed and empirically validated, the conceptual novelty is limited. The paper contributes mainly at the system level rather than proposing a fundamentally new learning principle or optimization mechanism.

2.Questionable personalization authenticity: The “similar user text” retrieved from the user–item graph provides co-occurrence and population-level preference signals rather than true individual-specific information. As such, the personalization improvement may stem more from global correlations (shared neighbor statistics) than from a deeper understanding of each user’s distinct style or intent. The paper does not include qualitative or quantitative analyses demonstrating that the generated content reflects unique user preferences instead of generic consensus patterns.

3.Potential noise amplification from graph augmentation: The link prediction model used for graph-based context expansion inevitably introduces spurious or noisy edges, especially in sparse or biased user–item graphs. These errors may propagate through the synthetic history and distort the personalized style, particularly when the predicted neighbors are semantically inconsistent.

**Questions:**

1.How do the authors ensure that the “similar user texts” extracted from the user–item graph provide personalized rather than population-level signals? Have the authors conducted any analyses (e.g., user-level style divergence, lexical overlap, or embedding similarity) to confirm that the generated outputs are truly distinctive to each user?
2.How sensitive the generation quality is to incorrect or noisy user–item edges?

---

> ### Author Response · Authors · 2025-11-21
> **Reply to Reviewer gV7p - Part 1**
>
> ### Weaknesses
> 1. We appreciate the reviewer’s comment and would like to clarify that GRASPER is not a simple combination of graph-based personalization and reasoning-enhanced generation, for two main reasons.
>     - First, while graph augmentation and reasoning have each been explored independently, we redesign both components to jointly address a sparsity-specific failure mode not covered in prior work. In Au et al. [1], the graph is used solely as a retrieval mechanism for existing neighboring reviews. In Salemi et al. (2025) [2], reasoning is used only for self-training to improve generation quality. Neither work constructs synthetic user–item interactions, nor aligns them via reasoning before they are incorporated into the user profile. GRASPER introduces a new paradigm—reasoning-aligned synthetic profile expansion—in which predicted interactions are instantiated as synthetic texts and then corrected through reasoning alignment before being used for personalization. This is a fundamentally different usage of both graph context and reasoning, and we believe it offers a new conceptual direction for the community.
>     - Second, we discover and formalize a new empirical/theoretical phenomenon that does not appear in prior graph-based personalization systems. As shown in Table 5 and Appendix F, GRASPER exhibits a unique trend where increasing the expansion size K consistently improves performance, because reasoning alignment actively reduces the stylistic mismatch (bias) introduced by predicted neighbors. In contrast, PGraph performance degrades as K grows, since adding more retrieved text introduces noise without any mechanism to correct it. We formalize this contrast as a bias–variance tradeoff, showing that graph retrieval alone is insufficient without reasoning-based alignment, and that the interaction between expansion and reasoning is what unlocks the performance gains.
>
>     We believe that these contributions go beyond the “A+B composition” and represent a new, integrated framework that reveals previously unobserved behaviors and introduces a novel paradigm for personalization under sparse user histories.
>
> 2. We thank the reviewer for pointing out the potential concern on the personalization authenticity. Our proposed GRASPER is designed to ensure that the outputs align with each user's individual style. First, when retrieving _similar user text_, we compute the similarity based on the style-aware encoder [3], which ensures that the retrieved users share stylistic characteristics rather than purely co-occurrence patterns. Second, the reasoning-alignment stage conditions the synthetic texts on the user's own writing, which explicitly calibrates the retrieved signals back towards the user's style, as we can see from $\mathcal{L}_\text{gen}=\text{CE}(\mathcal{M}(\rho(x)), t_{u, j})$ in Equation 9.
>
>     The case study in Figure 3 demonstrates that GRASPER is able to capture more nuanced stylistic characteristics of user's writing comparing to PGraph which only directly retrieves neighborhood text and generate without reasoning. For example, in the first example, GRASPER mirrors the user’s succinct, matter-of-fact style (“spray bottles broke in the first use”), whereas PGraph produces a generic and less grounded summary (“Disappointment”). Across all examples, GRASPER consistently preserves lexical choices, and stylistic patterns of the user’s writing, while PGraph’s outputs tend to reflect broader consensus trends.
>
> 3. We thank the reviewer for raising the important concern regarding the potential noise introduced by graph augmentation. To thoroughly examine the effect of such noise, we first analyze the link prediction performance. We then evaluate how this prediction quality influences the downstream reasoning-based personalization process, showing that our framework maintains stable behavior across varying levels of augmentation performance. Detailed experimental results and analyses are provided in our response to Question 2 (Q2).

---

> ### Author Response · Authors · 2025-11-21
> **Reply to Review gV7p - Part 2**
>
> ### Questions
> 1. As briefly mentioned above, to ensure that the extracted _similar user texts_ reflect personalization signals instead of population-level signals, we first utilize a style-aware encoder [3] to compute the similarity. Additionally, the reasoning alignment outlined in Equation 9 conditions the text generation on user's own writing, which regulates the generation to model user's styles. To further demonstrate that the retrieved _similar user texts_ can provide personalization signals, we compute the cosine similarity between the embeddings of the target user's text and the retrieved user's text, as well as the target user's text and random reviews on the Amazon dataset. We report the result below:
>
> | Comparison Type                          | Mean Cosine Similarity |
> |------------------------------------------|-------------------------|
> | User vs. Similar-User Neighbors          | 0.3343              |
> | User vs. Random Review                   | 0.1601             |
>
> This shows that retrieved similar-user texts are much closer to the target user's own writing than random population reviews.
>
> Moreover, to show that the generated outputs are distinctive to the target user, we perform an authorship attribution test on the generated content. For each generated review, we compare its embedding similarity to the target user's style vs. 10 randomly sampled other users.
>
> | Settings| Value   |
> |---------------------------------------------|:-------:|
> | Random      | 0.55 |
> | Target   | 0.67 |
>
> The similarity for the target user is far above the random chance level of 0.0909, indicating that the generated outputs are distinctive to each user's style.
>
> 2. We thank the reviewer for the question regarding how sensitive our generation quality is to noisy user–item edges. To directly evaluate this, we first report the link prediction performance of our retrieval module, which remains strong across datasets. However, we acknowledge that noise will inevitably be injected during the process. We then examine whether the noise affects downstream personalization by splitting all test instances into the top and bottom 50% based on their link-prediction confidence score. As shown in the following table, the generation quality of the low-confidence group remains comparable to the high-confidence group across metrics. This demonstrates that the reasoning module can still extract stylistically relevant signals from the retrieved neighborhood even with mixed signals.
>
> | Dataset   | MRR   | Hits@1 | Hits@5 | Hits@10 |
> |-----------|-------|--------|--------|---------|
> | Amazon    | 0.531 | 0.415  | 0.659  | 0.760   |
> | Hotel     | 0.324 | 0.210  | 0.446  | 0.546   |
> | Feedbacks | 0.275 | 0.178  | 0.394  | 0.477   |
>
> | Group                   | ROUGE-L Mean | METEOR Mean |
> |-------------------------|:-----:|:------------:|
> | Bottom 50% link pred scores      |   0.587982   |  0.555729   |
> | Top 50% link pred scores  |   0.605974   |  0.569973   |
>
> **Note**: All the additional experiments and content has been added to the latest version of the paper PDF.
>
> [1] Personalized Graph-Based Retrieval for Large Language Models. Au, et al. 2025.
> [2] Reasoning-Enhanced Self-Training for Long-Form Personalized Text Generation. Salemi, et al. 2025.
> [3] Same Author or Just Same Topic? Towards Content-Independent Style Representations. Wegmann, et al. 2022.

---

### Author Response · Authors · 2025-11-28
**Gentle Follow Up**

Dear Reviewers,

As we enter the final week of the discussion period, we would like to gently follow up. Our author response has been posted, and we remain available to address any further questions or clarifications. We appreciate your time and would welcome any additional feedback.

Best regards,

Authors

---

### Meta-Review · Area_Chair_kCqV · 2025-12-28

**Summary:**

This work presents the GRASPER framework that can be used to enhance personalized text generatio for users with limited historical data. The framework introduces several ideas, including graph based context expansion to enrich sparse user histories via link prediction and a reasoning aligned generation to generate personalized content via generation of an explicit reasoning path to align synthetic and real user data. The authors conduct evaluation of their framework on a variety of domains.
One of the main concerns raised by reviewers in the evaluation of this work is the limited scope of the contribution. In particular, it was argued that many of the components in the proposed framework, such as reasoning trace modeling, memory based personalization and others are components adapted from previous works.
The empirical evaluation is restricted to few shot personalization scenarios, making it unclear how this framework would work in long-horizon or multi-user settings. Additionally, there are missing personalization benchmarks in the empirical evaluation of this work such as LAMP. Unfortunately because of these concerns I cannot recommend acceptance.

**Reviewer Concerns:**

The main concern, echoed by a few reviewers is the limited scope of the conceptual novelty of this work. It is a very well executed work that combines a few existing ideas together. Additionally, some concerns were raised in the evaluation of the proposed methods,  including the lack of certain benchmarks.

**Reviewer Scores:**

The reviewers failed to engage in the discussion at the time when it was stopped, making me believe they would not have changed their scores much.

---

### Decision · Program_Chairs · 2026-01-26

Reject